



# Imaging and quantification of the pore microstructure of gas shales using X-ray microtomography

*Mozhdeh Mehrabi[1], Mehrdad Pasha[1], Ali Hassanpour[1*], Paul W.J. Glover[2] & Xiaodong Jia[1]*

[1]Institue of Particle Science and Engineering, School of Chemical and Process Engineering, University of Leeds, Leeds, LS2 9 JT, UK
[2]School of Earth and Environment, University of Leeds, Leeds, LS2 9JT, UK
*Corresponding author email, A.Hassanpour@leeds.ac.uk

## Abstract

Optimisation of gas production from shale gas reservoirs depends critically upon a good understanding of the porosity and pore microstructure of the shale. Conventionally surface area measurements or mercury porosimetry have been used to measure the porosity in gas shales. However, these conventional methods have limited accuracy and only provide a bulk measurement for the samples. More recently, scanning electron micrography (SEM) and Focussed Ion Beam SEM (FIB-SEM) techniques have been applied in an attempt to address these limitations. Unfortunately, these two methods destroy the samples. In this research three-dimensional x-ray micro tomography (XRMT) imaging techniques were used to capture the structure of three samples and also compared to data from mercury porisimetry. The resulting data have been segmented in order to recognize individual pores down to a resolution of about 1 μm. Distributions of pore volume, pore size, pore aspect ratio, surface area to pore volume ratios and pore orientations were calculated from the XRMT data. It was found that the porosity obtained from XRMT measurements is smaller than that obtained using mercury porisimetry, the reason for which might be displacement of kerogen by the high pressures generated in the mercury technique, but is unlikely to be due to both techniques not being able to measure pores smaller that about 900 nm. Pore volume and size distributions showed all of the shales tested in this work to be multimodal with similar major modal values for volume and pore size. The pores also have a range of pore aspect ratios and surface area to pore volumes, including values indicating the presence of significant oblate spheroidal pores where the major axis is up to 330 times bigger than the minor axis. This has implications both for the connectedness of pores and the resultant gas permeability and the effectiveness of gas desorption processes into the gas shale's pores. These high aspect ratio pores were oriented both in dip and azimuth in preferential directions making it likely that the shale gas itself has significant anisotropy both for permeability and in its mechanical properties. Permeabilities calculated from the XRMT distribution data matched very well with permeabilities obtained by scaling considerations and typical values for similar gas shales.

**Keywords.** Gas shales, XRMT, anisotropy, porosity, permeability, heterogeneity, mercury porosimetry.



## 1.  Introduction

Recently research into extracting unconventional resources has increased as oil and gas
production from conventional reservoirs continues to decline. Within fifty years it is
expected that all hydrocarbon reservoirs will either be small, low permeability,
heterogeneous, anisotropic, found in difficult to reach locations, or some combination of
these (Miller et al., 2014). Consequently, unconventional reservoirs are becoming an
important alternative source of natural gas to meet the huge global demand for energy
(Alfred and Vernik, 2012).
According to IHS Markit (Edwards, 2015), unconventional reservoirs already account
for about two thirds of current global reserves. However, the extraction of hydrocarbons
from these extremely low porosity and permeability rocks is extremely difficult. Not only do
we not know how much of it we might be able to extract, its extraction requires the use of
new techniques and special recovery operations whose cost makes producing these
reservoirs often marginally economic and less hydrocarbon prices rise.
Unconventional hydrocarbon resources exist in a number of different forms
including tight gas and oil reservoirs, coal bed methane deposits and thick formations
containing shale gas. This paper focuses on significant shale gas deposits. It recognises that
the gas held in shale gas deposits occupies a pore microstructure of which little is known, at
least at a microscopic scale. Since it is these rocks that need to be hydraulically fractured so
that gas will flow from them, we consider that a better understanding of the microstructure
of gas shales will be extremely useful in designing ways to extract more shale gas from them.
According to the United States Energy Information Administration (USEIA), 60% of
the Earth's sedimentary crust consists of shale, and the organic matter in it is the primary
source of all hydrocarbons, as either a gas or oil (Blyth and De Freitas, 1984). Shale is a fine
grain sedimentary rock derived from clastic sources and which contains a significant amount
of different clays mixed with fragments of quartz and other minerals. The organic material
that is deposited with these mineral particles (clasts) is altered by temperature and pressure
(Tissot and Welte, 1978) leading to the formation of kerogen and the creation of
maturation-induced pore space filled with hydrocarbons. However, these hydrocarbons
remain trapped within the shale because of the rock's ultra-low permeability (Alfred and
Vernik, 2012). The increased pore pressure created through hydrocarbon generation could
also result in maturation-induced micro-cracks (Vernik and Liu, 1997) that may provide
increased migration of hydrocarbon into reservoirs, creating the world's conventional oil and
gas resources. However, much of the hydrocarbons, remain in the shale source rocks
because they occupy and cannot leave a rock microstructure consisting of millions of
extremely small and often unconnected pores (Alfred and Vernik, 2013). During production,
access to this trapped gas is currently improved by the use of hydraulic fracturing. However,
this process is presently a very much hit and miss affair because we do not know how the
shale gas is distributed within the gas shale at a microscopic to macroscopic scale
(Richardson et al., 2013) and we have limited ability to control and focus the growth of
fractures.
Improvements to hydraulic fracturing design and proppant technology have already
led to a step change in shale gas production rates. Significant improvements have been
shown to occur when the hydraulic fracturing takes full account of the mechanical
properties of the rocks (Glover et al., 2000), while high aspect ratio fibre and channelized
proppant technology (Schlumberger, 2015) has produced up to 20% greater production
rates. Further increases in production rates are likely to be possible by designing hydraulic
fracturing campaigns that take account of the microscopic distribution of the hydrocarbons
within the shale, but for this to be done we also need more information about how the
hydrocarbons are distributed within the shale at a microstructural level (Gerke et al., 2013).
Up until now, most studies of the pore structure of shale have used mercury
injection capillary pressure (MICP) and nuclear magnetic resonance (NMR) measurements



(Sondergeld *et al.*, 2010). The mercury technique, though extremely useful in conventional
reservoirs, is less relevant in shales since the injection pressures need to be extremely high
to mercury penetrate into the rock at all. These higher pressures begin to compress the rock,
crushing the pore spaces that the technique is supposed to be probing, and leading to
overestimation of capillary pressures together with underestimations of pore size, pore
throat size and porosity. Additionally, neither of these techniques provides information
about the microstructure of the pores and how they are connected. On the other hand the
NMR technique provides some information about the microstructure but suffers from low
resolution and cannot measure the connectivity of the pores.
Consequently, another approach is needed. The microstructure of shale has been
imaged extensively using Focused Ion Beam (FIB) SEM techniques (Chalmers *et al.*, 2012;
Loucks *et al.*, 2009; Ambrose *et al.*, 2010; Passey *et al.*, 2010; Schieber, 2010; Sondergeld *et
al.*, 2010). However, use of the FIB-SEM method to characterize the 3D microstructure of
rock is destructive and very time consuming. A better alternative for quantifying pore
structure would be to use 3D X-ray tomography because it is non-destructive, fast, and
allows the same sample to be scanned repeatedly. Other allied technologies such as NMR
scanning (Sondergeld *et al.*, 2010) and Positron Emission Tomography (Ogilvie *et al.*, 2001)
suffer from the same low resolution (approximately 1 mm). We are beginning, however, to
see the use of X-ray micro-tomography (XRMT) (Iglauer *et al.*, 2013; Panahi *et al.*, 2014;
Reipe *et al.*, 2011; Mayo *et al.*, 2015; Peng *et al.*, 2012; 2015) using highly focused X-Ray
beams in the laboratory, which can attain resolutions better than 1 micron. Standard X-ray
micro-tomography apparatus can attain resolutions down to 760 nm in ideal conditions,
which is sufficient to image most pores in shale, while some apparatus can provide
resolutions as low as 20 nm.
This work describes an X-ray micro-tomography study to image the microstructure
of samples of gas shale at a micron-scale in order to characterise the pore structure.
Information has been gathered on microstructural parameters such as the location, size,
volume, shape, surface area to volume ratio and preferred orientations of pores in order to
help understand how the rock was formed, how it acts as a reservoir for gas, how we can
improve gas permeability in such rocks, and how, ultimately, we can extract more gas in an
efficient manner. We believe that the characterization of gas shale pore structure must lead
to improvements in the amount of gas we can extract from a given reservoir.
X-ray microtomography like other techniques has some limitations (Blunt et al.,
2013), including resolution limits, a trade-off between resolution and sample size, and
difficulties in segmentation for materials (or phases) with similar densities due to similar X-
ray absorption coefficients. However, these limits are more than made up for by the
advantage is the technique has over other imaging techniques when it comes to the imaging
of gas shales.
In this work we have recognized that before characterisation of gas shales requires
an even higher resolution than the 900 nm resolution reported here, and we are currently
carrying out further imaging with a much better resolution (down to 50 nm), which will be
the subject of further publication.

## 2.   Methodology
### 2.1 Samples
The samples imaged in this work have a European source, but due to a confidentiality
agreement, it is not possible to disclose further details. Associated mercury injection
capillary pressure (MICP) measurements have indicated that the porosity varies between
2.8% and 10.4%, while X-ray diffraction (XRD) measurements have shown that the samples





are composed of 49.2 and 58.3 wt.% clays, 24.2 to 29.4 wt.% quartz and feldspars, 3.7to 16.1
wt.% carbonates and 2.5 to 8.9 wt.% kerogen (Table 1).

### 2.2 Sample preparation

In order to optimise the scanning resolution, the sample should be as small as possible and
the X-ray source should be brought close to the rotating sample.
In this work samples were prepared by taking a small core of shale and cutting it into
2 pieces of about 2×1 cm each. Each piece was then mounted on a glass slide (48×26 mm)
using thermoplastic wax. In order that the data from different samples can be compared our
preparation protocol demands samples are cut into approximately the same size. Each face
of the mounted sample was first machined to 1 mm thick using a Buehler PetroThin
instrument, turning the sample over and remounting it on the glass slide until a cube of side
1 mm was all that remained. Once complete a similar process was carried out on other faces
to reduce the sample to a cube of about 500 μm in all dimensions. The thickness of wax
between the glass slide and the surface of the sample has been estimated to be 15-20 μm,
leading to the corresponding uncertainty in sample size. The samples were finally cleaned
with acetone and mounted at the top of a rotating sample holder with cyanoacrylate epoxy.

### 2.3 X-ray Microtomography

X-ray micro tomography (XMT) is a non-destructive, relatively fast and accurate technique,
which can reveal the internal structure of the shale samples. The technique can be used to
scan the sample as many times as needed to visualize internal properties and build a 3D
internal structure of the samples (Bakke and Oren, 1997; Li et al., 20010; Curtis et al., 2010;
Gelb et al., 2011).
The process of X-ray computed tomography (XRMT) consists of taking a number of
X-ray radiographs (referred to as projection images) at various angles by projecting an X-ray
beam through the specimen and measuring the attenuation of the beam received on a
detector (Markowicz, 1993), as shown in Figure 1. Attenuation is quantified in terms of a CT
number, with a larger CT numbers being associated with materials have a higher atomic
number and density. The projection images are obtained at a large number of different
angles as the sample rotates. A technique known as Filter Back-projection (Mersereau and
Dudgeon, 1975) can then be used to reconstruct the 3D volume of the specimen.

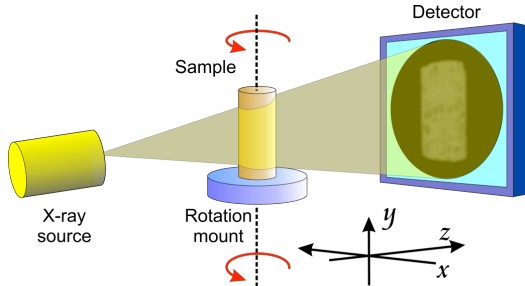

**Figure 1. Schematic representation of a computerised micro-tomography measurement set-up.**

X-ray Microtomography (XRMT) has enough sensitivity to distinguish gas-filled pores
from solid kerogen primarily due the large difference in their densities, which leads to a
contrast in their CT numbers. In our research the contrast between kerogen and gas-filled
pores has been enhanced by adjusting the X-Ray power (*i.e.*, the voltage and current).  A
simple method for checking whether a pore of crack is gas-filled is to compare the grey-level



of the pore or crack with voxels outside the sample. Kerogen mapping will be the subject of
a future publication.
189   Once the 3D volume of the specimen has been obtained, a series of image analysis
techniques can be used to visualise the internal structure of the specimen and obtain digital
information on its 3D geometry and structural properties.
192   In this study a GE Phoenix Nanotom (XRMT) instrument at the Institute of Particle
Science and Engineering at the University of Leeds has been used to obtain the 3D volume of
the samples. This apparatus has a microfocus X-ray generator and narrow beam, which
allows for the examination of high-density materials such as rocks. The final resolution is
determined by the sample size, beam quality and the detector specifications as well as the
position of the rotating sample with respect to the beam and the detector. For the samples
studied in this research the voxel resolutions of the images were 1.2 μm for Sample 1, 0.9
μm for Sample 2, and 1.0 μm for Sample 3.
200   VGStudio software was used to reconstruct the images from projection images and
Avizo Fire software was used for image analysis on the obtained volumes. The image analysis
provided sample porosity, pore volume, pore aspect ratio, the ratio of the pore surface area
to pore volume, the distribution of pore throat sizes, the connectivity of the pores and any
preferential directionality (anisotropy) in the pore distribution.
**2.4 2D SEM Scoping Study**
To complement this study, samples were also investigated using a scanning electron
microscope (SEM) and associated energy dispersive spectroscopy (EDS) imaging. The data
acquired in these experiments, allowed identification of the type of minerals and the
presence of pre-existing cracks. For example, bright spots in images, often composed of
clusters of crystals, as shown in all three panels of Figure 2 indicate pyrite framboids, while
fractures are visible clearly.

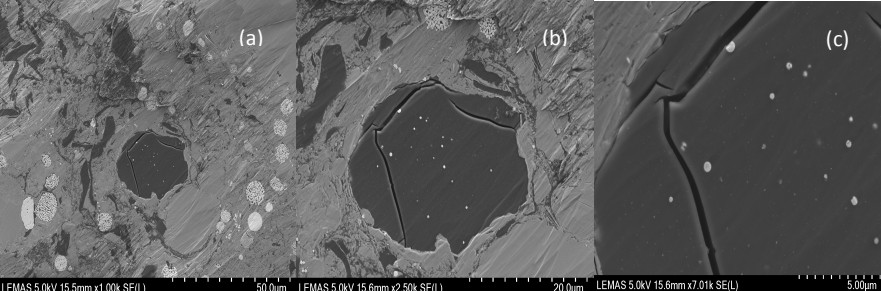

**Figure 2. SEM image of Sample 2 at (a) 50 μm, (b) 20 μm and (c) 5 μm resolutions. The lightest areas**
**correspond to dense material with high atomic number such as pyrite, the darker areas represent low density,**
**and low atomic number components such as organic materials, and the darkest regions indicate pores and**
**cracks.**
**3.   Results and Discussion**
**3.1 Numerical analysis of scan data**
Figure 3 shows the reconstructed and filtered three-dimensional images from Sample 1.
Figure 3(a) shows clearly the complex nature of the microstructure of the shale. There are
connected and unconnected pores at all scales and of all aspect ratios as well as pre-existing
fractures, again at all scales, some of which may have been the result of sample preparation.
In addition there is a complex mixture of minerals including high-density pyrite, which has a
high CT number, and shows up at small white small aspect ratio spots.
227   The pores were segmented using a defined range of grey values corresponding to
gas-filled pores by using manual thresholding, as shown in Figure 3 (b). It is worthwhile



noting that most of the pore space is not well-represented in this figure due to the
resolution of the figure rather than the resolution of the data.
Figure 3(c) and (d) show the three-dimensional pore structures of Sample 1, the
pores have been colour coded according to whether contiguous voxels are part of the same
pore. In this way each colour represents a fully connected pore. (It should be noted that two
clearly separate patches that have the same colour are not connected, but share the same
colour simply because of cycling over a limited number of colours in the available palette.)
It is possible to analyze the size and spatial distribution of the pore space as well as
its connectedness.

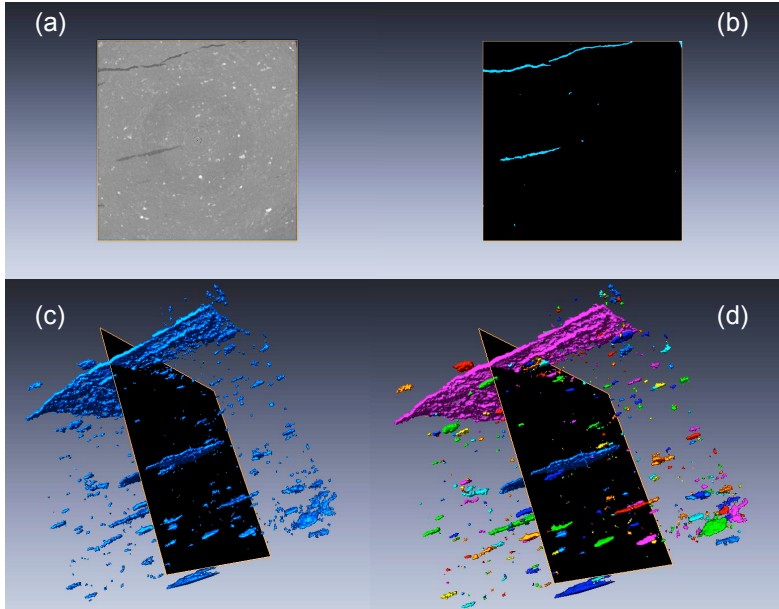

**Figure 3. Image processing workflow, (a) 2D slice of a 0.5 × 0.5 ×0.5 mm volume of Sample 1 using a non-local**
**mean filter, (b) the segmented of pore spaces obtained by thresholding with specific range of CT numbers**
**(represented by grey-levels) corresponding to pores, (c) 3D volume of pores for the Sample 1, (d) 3D image of**
**the connected pores, as the cluster of connected pores are shown in same colour.**

Table 1 shows a selection of the most important data from the analysis of the three
samples. The most obvious conclusion from the data in Table 1 is that the porosity derived
from X-ray micro-tomography is significantly (between 1.5 and 10 times) less than that
provided by MICP measurements. This discrepancyis difficult to explain by experimental
inaccuracies, and leads to questions over whether use of either the porosity from the MICP
technique or the porosity calculated from the micro-tomography is correct in gas shale. One
possible cause of the discrepancy is that the fact that the MICP measurement is
overestimated due to the high pressures damaging the sample. However, one would expect
this to reduce the measured porosity rather than increasing it. Another explanation might be
that XRMT at the resolutions available to us are not taking into account pores smaller than
our resolution limit (about 900 nm), which would imply that nanopores are extremely
important in gas shales. However, these small pores should also be missed by the MICP
measurement. Alfred and Vernik (2012; 2013) have recently published a new petrophysical
model for gas shales, distinguishing between open porosity and kerogen-filled porosity.
Consequently, another source of the apparent discrepancy between the two porosities
would arise if the high pressures used in the MICP technique have disturbed the kerogen in
the rock samples. This would lead to the MICP measurement returning a porosity composed
of the initial gas-filled porosity and some of the space previously occupied by kerogen,



leading to an overestimation of the gas-filled porosity of the gas shale. Ward (2010) reported
that the density of kerogen in the Marcellus shale varies with thermal maturity in the range
1.53 to 1.79 g/cm$^3$. If we take a mean density of 1.65 g/cm$^3$ for kerogen and 2.7 g/cm$^3$ for
the other solid components of the rock, we obtain kerogen values of 3.93% by volume for
Sample 1 and 3 and 12.56% for Sample 2. Clearly, there is ample scope for the process
proposed by Alfred and Vernik to occur. If such a process does occur, it would be extremely
important to know what technique was used to measure the porosity of gas shale from a
hydrocarbon potential point of view. Furthermore, comparison of porosity measurements
using two different techniques would possibly allow the fraction of kerogen in the rock to be
determined.

| Parameter | Unit | Sample 1 | Sample 2 | Sample 3 |
|---|---|---|---|---|
| **Composition** | | | | |
| **Clays** | | 49.2 | 58.3 | 49.2 |
| **Quartz & feldspar** | w.t% | 29.4 | 24.2 | 29.4 |
| **Carbonate** | | 16.1 | 3.7 | 16.1 |
| **Kerogen** | | 2.5 | 8.9 | 2.5 |
| **Porosity from MIP** | (-) | 0.104 | 0.028 | 0.104 |
| **Pore voxels count** | (-) | 633 | 100 | 258 |
| **Total voxel count** | (-) | $10^9$ | $10^9$ | $10^9$ |
| **Spatial resolution (voxel size)** | (μm) | 1.2 | 0.9 | 1.0 |
| **Voxel volume** | (μm$^3$) | 1.73 | 0.73 | 0.82 |
| **Porosity from microtomography** | (-) | 0.0071 | 0.0029 | 0.0096 |
| **Volume of smallest pore** | (μm$^3$) | 13.8 | 2.19 | 4.6 |
| **Volume of largest pore** | (μm$^3$) | $5.97 \times 10^6$ | $9.96 \times 10^4$ | $2.47 \times 10^5$ |
| **Mean pore volume** | (μm$^3$) | $1.85 \times 10^4$ | $7.32 \times 10^3$ | $3.19 \times 10^3$ |
| **Median pore volume** | (μm$^3$) | $6.15 \times 10^2$ | $1.32 \times 10^3$ | $7.28 \times 10^2$ |
| **Effective pore radius** | (μm) | 1.43 | 1.33 | 1.42 |
| **Formation factor** | (-) | $2.8 \times 10^6$ | $41 \times 10^6$ | $1.13 \times 10^6$ |
| **Estimated permeability** | (nD) | 92.3 | 5.5 | 22.3 |

**Table 1. Porosity, pore microstructure and estimated permeability parameters associated with three samples**
**of shale gas measured in this work.**
3.2 Pore size and pore volume distributions
The pore volume distribution for each sample is shown in Figure 4 and can be seen to cover
an extremely wide range, from the resolution of the technique in Sample 2 (900 nm) to
about five orders of magnitude higher. Figure 4 shows both the incremental and cumulative
distribution of pore volumes for each of the three samples. It can be seen that pore volumes
range from below $2 \times 10^{-9}$ mm$^3$ to over $2 \times 10^{-4}$ mm$^3$. For these three samples all pore volume
distributions are multimodal but the largest contribution to pore volume in all three samples
occurs at a pore volume of about $1.5 \times 10^{-6}$ mm$^3$, accounting for about 18% of the total pore
volume for Sample 1, 21% for Sample 2 and 41% for Sample 3 by pore number count.
The lower limit of the distribution measured in this work is controlled by the
resolution of the technique, with samples 1 and 3 showing a marked reduction in measured
pores with volumes less than $1.5 \times 10^{-8}$ mm$^3$, and $2.5 \times 10^{-9}$ mm$^3$ for Sample 2. The multimodal
character of the distributions hints that there may be significant pore volume in the form of
pores with sizes less than the resolution of the XRMT technique.
The upper limit to pore sizes in Sample 2 and Sample 3 is about $1.5 \times 10^{-4}$ mm$^3$ and
$2.5 \times 10^{-4}$ mm$^3$, respectively, while for Sample 1, with the presence of a one large crack, it is
$6 \times 10^{-3}$ mm$^3$.



The cumulative distributions in Figure 4 show that Sample 1 and Sample 3 have
consistently higher pore volumes than Sample 2 despite the similarity apparent in their
associated incremental pore volume distributions.

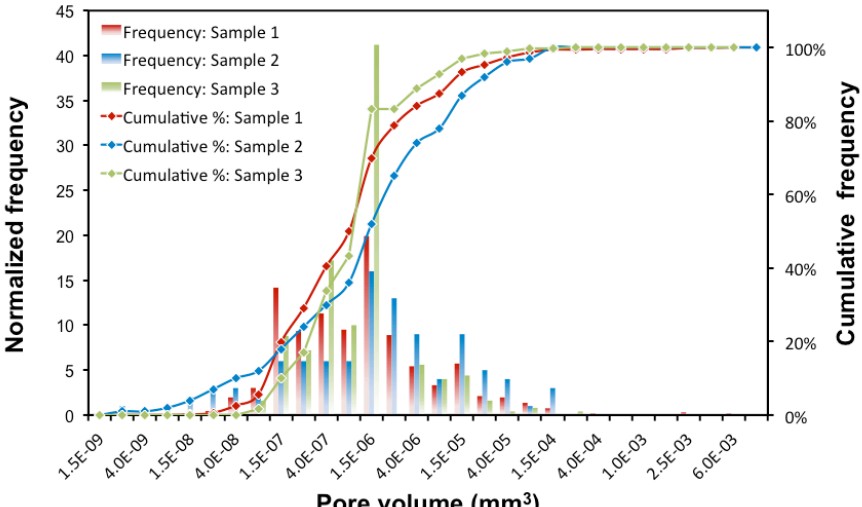

**Figure 4. Incremental and cumulative pore volume distribution for Sample 1 (Red), Sample 2 (Blue) and Sample**
**3 (Green).**

### 3.3 Pore aspect ratios

Pores can be considered to approximate to an ellipsoid with radii $a$, $b$ and $c$ in each of the
three orthogonal directions $x_1$, $y_1$ and $z_1$, where $x_1$ is taken along the maximum length of the
ellipsoid, $y_1$ along the next largest, and $z_1$ along the smallest ellipsoidal dimension. Aspect
ratios can then be described as the ratio of pairs of each of these orthogonal lengths. In this
work we calculate the aspect ratio of two largest bounding box dimension of pores, which is
given by the ratio $a/b$, where $a \geq b \geq c$. Feret's diameter (Merkus, 2009) was used to calculate
the values of $a$ and $b$ for each pore from the numerical data set.



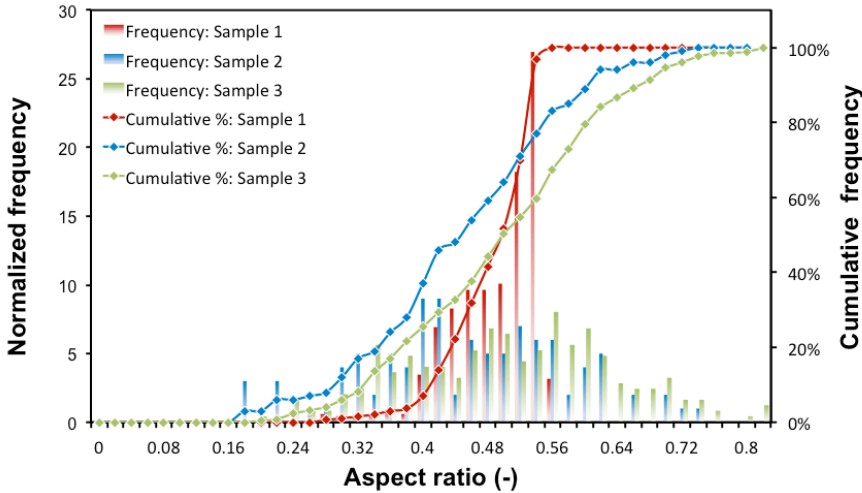

**Figure 5. Aspect ratio distributions for the shale gas samples for Sample 1 (Red), Sample 2 (Blue) and Sample 3 (Green).**

Pore aspect ratio is a very important parameter in the characterisation of gas shales because it is not only related to the connectedness of the pores (Glover, 2009), which influences the electrical and fluid transport properties of the rock, it is also related to how effectively matrix-bound and kerogen-bound gas can diffuse into the shales pore spaces. High aspect ratios provide more grain-to-grain contact, thus decreasing the pore compressibility (Saleh and Castagna, 2004). Although not evaluated in routine or special core analysis, the aspect ratio distribution of a rock affects the connectedness and tortuosity of pore spaces, which control formation factors, cementation exponents, saturation exponents and ultimately permeability.

Figure 5 shows that there is a well-defined preferred aspect ratio that is shared by all samples (0.54, 0.42 and 0.56 for Sample 1, 2 and 3, respectively). However, Sample 2 and 3 contain pores with a much wider distribution of pore aspect ratios than Sample 1, indicating that while Sample 2 and Sample 3 contain some pores which are almost spherical as well as others which are very crack-like, together with all shapes in between, Sample 1 contains only pores in the middle range, which are never near-spherical nor very crack-like. In fact, Sample 1 has a well-defined maximum pore aspect ratio of 0.56. The implication is clear; some gas shales contain more high aspect ratio pore spaces at a microscopic scale. These high aspect ratio pores are more likely to interlink and will be more likely to give these shales a larger natural permeability. Consequently, we ought to be searching for gas shales, which have high aspect ratios in order to take best advantage of any natural permeability that is present.

### 3.4 Pore surface area to volume ratio

The shape of each pore also affects its surface area to volume ratio, $\xi$. This ratio is important because large surface areas facilitate the diffusion of gas initially trapped in the matrix of the rock and in the kerogen into the pore spaces within the shale. This is a necessary step before hydraulic fracturing can open up access to these small pore spaces. A high surface area ensures that the diffusion process is more efficient, not only ensuring a good initial charge of gas in the micro-pores of the shale, but also allowing those pores to be recharged quickly once initial production has removed the initially accumulated gas.





341    Surface areas to volume ratios are best understood by assuming the ellipsoidal
pores to be spheroids of either oblate or prolate types. Oblate spheroids have semi-axis sizes
according to $a=b>c$, i.e., spheres squashed in the $c$-direction, and approximate to penny-
shaped cracks or pores. Prolate spheroids have semi-axis sizes conforming to $a>b=c$, i.e.,
spheres stretched in the $a$-direction, and approximate to needles. The volume of both types
of spheroid can be calculated using the formula

$$V = \tfrac{4}{3}\pi abc. \tag{1}$$
The surface area of the two types of spheroid differ slightly. They are

$$S_{oblate} = 2\pi a^2\left(1 + \tfrac{1-e^2}{e}\tanh^{-1}e\right), \text{ where }\quad e^2 = 1-\tfrac{c^2}{a^2}\quad\text{ and }\tag{2}$$

$$S_{prolate} = 2\pi a^2\left(1 + \tfrac{c}{ae}\sin^{-1}e\right), \text{ where }\quad e^2 = 1-\tfrac{a^2}{c^2}.\tag{3}$$
The surface area to volume ratio for each type are then

$$\xi_{oblate} = \tfrac{3}{2c}\left(1 + \tfrac{1-e^2}{e}\tanh^{-1}e\right), \text{ and }\tag{4}$$

$$\xi_{prolate} = \tfrac{3}{2c}\left(1 + \tfrac{c}{ae}\sin^{-1}e\right).\tag{5}$$

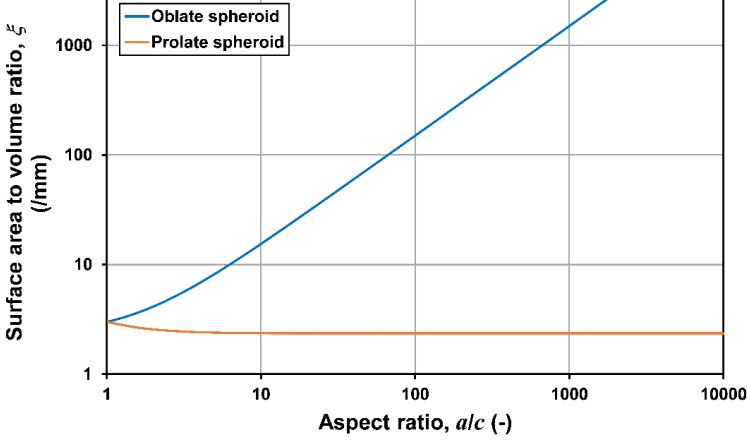

**Figure 6. Surface area to volume ratio as a function of aspect ratio for oblate and prolate spheroids, approximating to penny-shaped and needle-shaped pores, respectively.**

367    Figure 6 shows the relationship between the surface area to volume ratio $\xi$ of
oblate, penny-shaped pores and prolate, needle shaped pores to their respective aspect
ratios. It is clear that oblate pores provide a much greater surface area per volume than their
respective prolate pores. The XRMT data shows that the pores in the three samples we have
measured in this work are oblate with an aspect ratio of about 0.5, as shown in Figure 5.
However, Figure 5 shows that aspect ratios as high as 0.84 and as low as 0.16 are also
present.





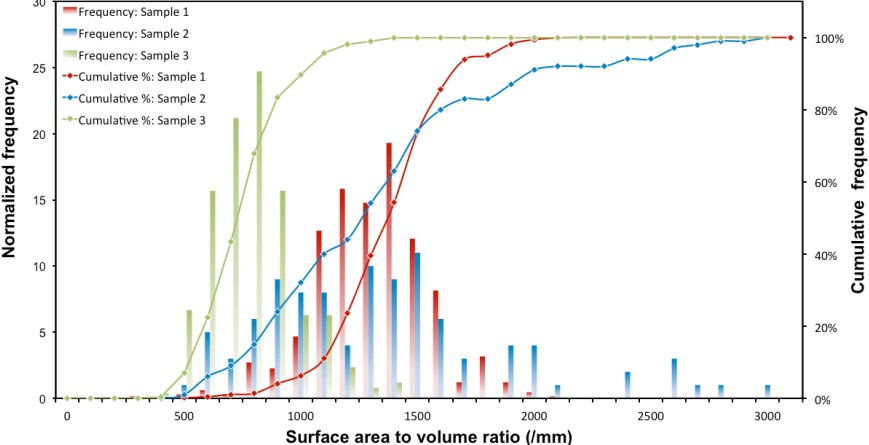

**Figure 7. Surface area to volume ratio distributions of the pores. Sample 1 (Red), Sample 2 (Blue) and Sample 3**
**(Green)**

Figure 7 shows the surface area to volume ($\xi$) distributions for the three samples
measured in this work. Samples 1 and 2 are similar with the $\xi$ ratio varying from about 500
/mm to values higher than 2000 /mm and similar modal values at about 1400±100 /mm and
1500±100 /mm, respectively. Sample 3 is clearly different, varying from about 500 /mm to
values no higher than 1400 /mm with a modal value at about 900±100 /mm.

The minimum surface area to volume ratios measured for all samples ($\xi$=500 /mm)
corresponds, according to the analysis in Figure 7, to an $a/b$ aspect ratio of about 330. In
other words, the penny-shaped pore is 330 times wider than it is thick. Likewise, the
maximum values of surface area to volume correspond to penny-shaped pores more than
1500 times wider than they are thick, with a modal behaviour for Sample 1 and Sample 2
showing $a/b$ aspect ratios of about 1000 and about 500 for Sample 3. The implication for gas
production is clear; Sample 1 and Sample 2 have twice the surface area than Sample 3 for
the diffusion of gas into the pores from the matrix, and these gas shales are likely to provide
better long-term resource than that represented by Sample 3 even though, the higher
porosity in Sample 3 will likely make it the better short-term prospect.
The surface area to volume ratio is also important in other respects. As indicated
previously, high aspect ratio and high surface area to volume pores are much more likely to
connect up with each other and therefore they are important in defining the natural
permeability of the shale. In this regard, Sample 1 and Sample 2 would be expected to have
a higher permeability than Sample 3. This is investigated later in this paper.
The shape of pores also is the importance in defining the geo-mechanical properties
of the rock. Shales have a tendency to plastic behaviour so any tendency to weakness is
likely to result in the closure of fractures and pores. High aspect ratio, high surface area to
volume penny-shaped pores and cracks are much more prone to closure than those with low
aspect ratios and low surface areas to volume (Glover et al., 2000; Curtis *et al.*, 2010).
Consequently, though high aspect ratios and high surface areas are beneficial for gas
production they are also likely to be found in shales which are difficult to produce from
because induced fractures will be more prone to closure in the long-term.
The previous analysis assumes that the pores behave like perfect smooth-surfaced
spheroids. Of course this is not the case in reality as can be readily seen in Figure 3. The



presence of rough surfaces on the pore walls increases the surface area to volume ratio
above that which would be expected by the overall aspect ratio of the pore. It has been
known for some years that not only pore size but also pore and fracture surfaces are fractal
(Nolte et al., 1989; Bahr, 1997; Ogilvie et al., 2006), and fractal pores can in principle have a
surface area to volume ratio that is infinite. Consequently, it should be considered that some
samples might have much higher surface areas due to the roughness and of their surfaces,
which do not increase the pore volumes but provide much larger pore surface areas.
Approaches that take into consideration the fractal distribution of properties such as
porosity and grain size are now being implemented in new reservoir modeling approaches
and used to create fractal permeability models for shale gas flow (e.g., Geng et al., 2016; Li
et al., 2016).

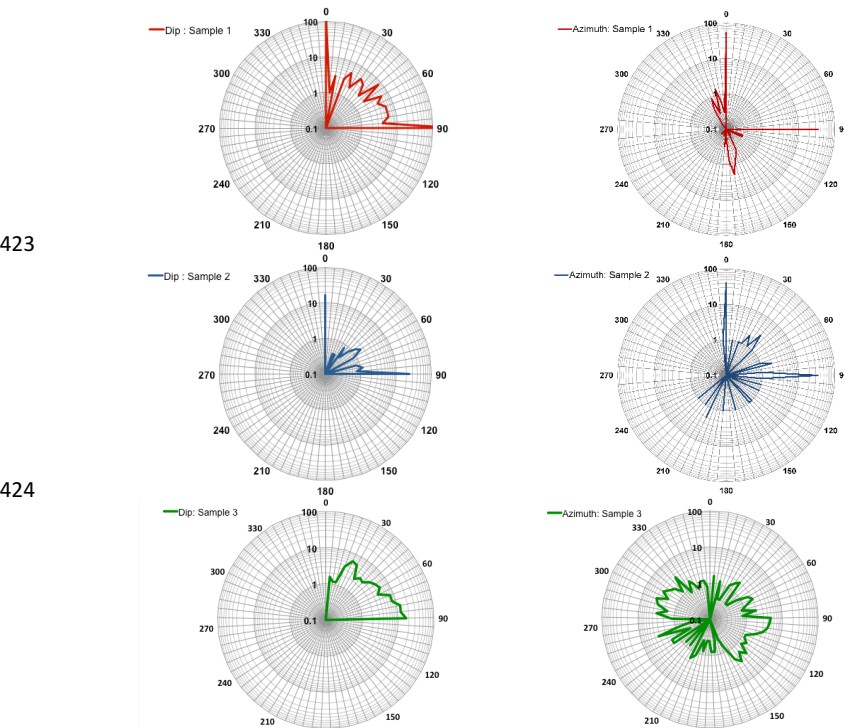



**Figure 8. Rose diagrams of Dip and Azimuth of the long axis of pores for each of the samples studied. Sample 1**
**(Red), Sample 2 (Blue) and Sample 3 (Green).**

**3.5 Pore orientation**
The XRMT data can also be analysed to ascertain the orientation of the pores according to a
polar co-ordinate system (Figure 8 shows the dip $\theta$ ($0 - 90^o$) and azimuth $\varphi$ ($0 - 360^o$) of the
major axis of the pores for each sample as a rose diagram. It is immediately clear that the
Sample 1 and Sample 2 are rather similar, showing marked preferential dips near $0^o$ and $90^o$,
which is parallel to the tomographic stage and also to the macroscopic bedding observed in
the samples. However, Sample 3 has preferential directions between $30^o$ and $80^o$. In
addition Sample 1 and Sample 2 show marked preferential azimuthal directions which are
orthogonal at $0^o$ and $90^o$ with additional secondary directions of which the two strongest are



170±5$^o$ and 34 5±5$^o$ for Sample 1 and 35±15$^o$ and 70± 3$^o$ for Sample 2. Overall it is Sample 1
that exhibits greater anisotropy, and this can be seen in the 3D image in Figure 3(c-d).

**3.6 Permeability**

The permeability of a rock may be estimated using the pore surface area to volume ratio ξ. It
is well known that the mean effective pore radius can be calculated using the (Johnson et al.
1986) approach, where the effective pore diameter $\Lambda = 2V_p/S_p$, where $V_p$ is the pore
volume and $S_p$ is the pore surface area. Consequently, $\Lambda = 2/\xi$. The $\Lambda$-value is a measure of
the aperture for fluid flow which controls the permeability of the sample according to the
relationship $k = \Lambda^2/8F$, where $F = \phi^{-m}$ is the formation factor of the rock (Glover, 2015).
In this equation the value of $\Lambda$ describes the size of the opening between the rock grains
allowing the passage of fluids, while the formation factor contains the information about
how connected or tortuous those fluid flow pathways are (Glover, 2009; 2010). The
formation factor was not measured directly in this work. However, since the cementation
exponent $m$ for shales varies between about 2.34 and about 4.17 (Revil and Cathles, 1999), it
is reasonable to assume a value of $m$=3. The formation factor can then be calculated using
the measured porosity for each sample.
The permeability for each sample can then be calculated, and is found to be 92.3 nD,
5.49 nD and 22.3 nD for samples 1, 2 and 3, respectively (Table 1), which is in agreement
with recent up-scaled permeability determinations for the Barnett shale (Peng et al., 2015).
It is worth noting that Sample 3 does not have a larger permeability than Sample 1 despite
having a larger porosity, which we ascribe to Sample 3 having a smaller surface area to
volume ratio that has not been compensated for completely by the larger porosity of Sample
3.
The dimensions of the interconnected pores have a major role in our estimation of
permeability and hence a viable theoretical method to find out effective pore radius or the
size of opening between the rock grains is required. In order to validate the previous
permeability calculations, ImageJ software has been used on SEM images of Sample 2 to
measure the equivalent circular diameter of a crack, which is similar to the measured mean
effective pore radius of same sample.
Figure 10 shows an SEM image of Sample 2 with a large crack. The crack has a length
of approximately 19 μm, and is approximately 0.3 to 0.5 μm wide. The equivalent diameter
(Jennings et al., 1988) of that crack has been calculated with the following equation, and the
value can then be compared with the effective pore radius of Sample 2 from Table 1.

$$d_e = 1.3(ab)^{0.625}/(a + b)^{0.25} \ ,$$

where; $d_e$ is equivalent diameter (μm), $a$ is the length of crack (μm), and $b$ is the width of
crack (μm). For the crack shown in Figure 10 the equivalent diameter is roughly equal to 2.53
μm and the effective radius of it is equal to 1.26 μm, which corresponds extremely well to
the effective pore radius of Sample 2 in Table 1 (1.33 μm).
It is interesting to note that, for porosity and hence, permeability measurement, the
equivalent diameter of cracks is not only depended on dimension of crack but also on flow
properties. So the concept of equivalent diameter was only expressed for comparative
purpose with effective pore radius.



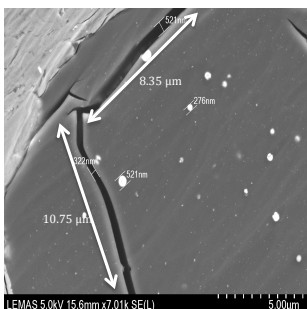

**Figure 9, SEM image of Sample 2 with dimension of crack**

## Conclusions

X-ray micro-tomography imaging (XRMT) has been used for qualitative and quantitative analysis of the pore structure of gas shale samples, attaining a spatial resolution of 0.9 to 1.2 μm. Pore structure can be determined easily using the X-ray tomography technique thanks to the large density contrast between the solid matrix and the pore fluid.

The distribution of pore volume showed a great variability of pore scales for all three samples, and different porosities (0.71%, 0.29% and 0.96% for Sample 1, Sample 2 and Sample 3 respectively). These porosities were significantly lower than those obtained on the same samples by MICP measurement. The probable reason for this discrepancy is that the micro-tomography is not taking account of pores on a nanometric scale. Sample 1 was found to have a narrow range of pore aspect ratios, centred on 0.55, with the pores being well aligned in a preferential direction, parallel to the bedding, while Sample 2 and 3 have a much wider range of aspect ratios, encompassing near-spherical pores and thin cracks, centred on 0.6, *i.e.*, close to the value for Sample 1. By contrast Sample 3 showed a less clear orientation of the pores. The surface area to volume ratio and permeability were calculated for all three samples.

Consideration of the porosity, pore size distributions, pore aspect ratio distributions, pore orientations and surface area to volume ratios as well as the calculated permeabilities shows Sample 1 to be the shale with the most shale gas potential.

Shale contains a wide range of pore sizes ranging from hundreds of microns down to a few nanometers (Alfred and Vernik, 2012). The lowest resolution achievable with the X-ray micro-tomographic instrument we used was about one micron. Hence, we expect that we have been analysing only the larger scale subset of the pores in the shale. Measurements are currently underway using an instrument with a nanometer scale resolution in order to ensure all sizes of pores are included in the measurement, and to examine whether the nanometer-scale pores are critical to our understanding of the pore microstructure of gas shale.

## Acknowledgments

The authors would like to thank Rodrigo Guadarrama Lara for his great assistance during the use of the X-ray micro-tomography device at Leeds University and the University of Leeds for providing Paul Glover with an academic dowry. We would also like to thank Harri Wyn Williams for his help of sample preparation in Earth and Environment rock preparation laboratory.

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
