# Peer review of "Imaging and quantification of the pore microstructure of gas shales using X-ray microtomography"

_Solid Earth, 2017_

## Referee Comment (RC1) · A. Aplin (Referee) · 14 Jun 2017

The 3D nature of shale porosity is not well established but is important for a range of problems related to fluid flow in the context of retention of hydrocarbons/CO2/nuclear waste in the subsurface, and for the storage and production of oil and gas from unconventional reservoirs. This paper uses X-ray microtomography as a tool to image the porosity of three shales. Unfortunately there is little or no geological context for the samples, which makes it more difficult to make a detailed analysis of the results. I don't really understand why it's not possible to say which geological formation the samples come from, although that isn't critical. More important is the fact that we don't know

the maturity of the samples and whether, for example, they are marine or lacustrine. The data would also be much more meaningful if there was a more robust petrographic characterisation of the samples. Indeed, microCT, given its resolution, is one way of looking at shale texture. The critical issue is what is being imaged and whether it is real or artefact. There is a significant literature now concerning the nature of pore systems in fine-grained sediments, much of which is not referenced in this paper. The reported XRD data indicate that the three samples are dominated by clay. Previous studies show that pores associated with clay matrix are generally well below the resolution of microCT, typically below say 20 nm. SEM and gas sorption studies both suggest that pores in organic matter are also, for the most part, very small, with only a small fractional volume above say 200 nm. Immature and oil window organic matter has essentially no macropores. Micron-size pores could be associated with thin silt layers within clastic sediments or with microfossils in more carbonate-rich zones. Are these relevant in this study? The segmented porosity shown in Figure 3 suggests that at least some of the porosity relates to microfractures and similar stress-relief/dessication features. Whilst commented on, there isn't a thorough and robust discussion of this critical issue. Without this, I fear that the results have limited geological and petrophysical meaning. I have made detailed comments and asked questions on the marked pdf, uploaded. But more generally, the paper needs more detail about: (a) the samples and their geological/petrophysical characterisation: maturity, texture, organic matter type (b) the image analysis process. Issues of segmentation, mixed signals from minerals and small pores, organic matter etc.. Uncertainties? (c) Comparison with MICP data. You use it only for total porosity (presumably injected porosity, assuming no compression of the samples?). Why not show the pore size data and use that in the discussion? As a final point, in both the introduction and in the discussion, I suggest that the paper should focus on the use of microCT as a tool for investigating porosity. Given the resolution of the technique and thus the uncertainties of what the data mean, the implications of the results for shale resource exploitation are unclear, for example in terms of permeability, diffusion, storage. The key issue, I believe, is what microCT can achieve

in terms of shale characterisation. That is likely to be textural assessments on scales of say 10 microns to a few millimetres.

Andrew Aplin 14th June 2017

Please also note the supplement to this comment:
http://www.solid-earth-discuss.net/se-2017-52/se-2017-52-RC1-supplement.zip

---

## Referee Comment (RC2) · Anonymous Referee #2 · 2 Aug 2017

This manuscript presents a study using micro-CT to characterize the pore structure in shale. Without finishing reading this manuscript, I have to reject it after reading Figure 3 because the segmented "pores" in this figure are either artificial cracks induced in sample retrieval or preparation, or layers of organic matter, or some low density features (that may or may not include pores). It just cannot be real pores. There are significant amount of literature focusing on pore characterization using SEM, FIB/SEM, nano-CT, or MICP, or nitrogen adsorption, and it is consensus that pores in shale matrix are mostly in sub-um scale, with small amount larger than 200-300 nm, if any. Instead, a substantial amount of pores are tens of nm to sub-nm-scale. The results and discussion based on the wrong interpretation of the micro-CT images are there-

fore meaningless.

Specific comments:

Ln 19: "Unfortunately, these two methods destroy the samples": the authors should realize that all methods destroy samples in some extent. Therefore, this is not a main reason for not using a specific technique.

Ln 27: "...but is unlikely to be due to both techniques not being able to measure pores smaller that about 900 nm.": I don't see the logic here. Micro-CT cannot measure < 1 um in this study, at the same time, MICP can measure all pores connected through > 3 nm pore throats. This will naturally make the micro-CT porosity (if doable) smaller than MICP porosity. Instead, "displacement of kerogen by the high pressures..." is only a speculation without data support.

Ln 32: "major axis is up to 330 times bigger than the minor axis.": As mentioned earlier, these cannot be pores.

Ln 120: "760 nm ....is sufficient to image most pores in shale": this is simply wrong. Most pores in shale is smaller than 100-200 nm, with a big amount of them smaller than 10 nm or smaller.

Ln 213, Figure 2. The resolution in Figure 2 is not 50 um, 20 um, and 5 um for a, b, and c, respectively. They are the bar length. The actual resolution, for example, in Figure 2c would be nm-scale. The caption of this figure is therefore wrong and misleading.

Ln 229: "that most of the pore space is not well-represented in this figure due to the resolution of the figure rather than the resolution of the data.": this statement indicates that the authors realized the resolution is not adequate. But unfortunately they continued to build the study based on something wrong. The statement of "due to resolution of the figure rather than the resolution of the data" is difficult to understand. What is the difference between resolution of the figure and the resolution of the data?

Typos:

[Figure]

Although I only read the first few pages, I found several typos:

Ln 7: "institue" → institute

Ln 21: "porisimetry" → porosimetry

Ln 88: "step" → steep?

Ln 247: "discrepancyis"→ discrepancy is